# Acute Kidney Injury and BK Polyomavirus in Urine Sediment Cells

**DOI:** 10.3390/ijms242417511

**Published:** 2023-12-15

**Authors:** Sahra Pajenda, Daniela Anna Gerges, Raimundo Freire, Ludwig Wagner, Zsofia Hevesi, Monika Aiad, Michael Eder, Alice Schmidt, Wolfgang Winnicki, Farsad Alexander Eskandary

**Affiliations:** 1Department of Internal Medicine III, Division of Nephrology and Dialysis, Medical University of Vienna, 1090 Vienna, Austria; sahra.pajenda@meduniwien.ac.at (S.P.); ludwig.wagner@meduniwien.ac.at (L.W.); n01618704@students.meduniwien.ac.at (M.A.); michael.eder@meduniwien.ac.at (M.E.); alice.schmidt@meduniwien.ac.at (A.S.); wolfgang.winnicki@meduniwien.ac.at (W.W.); farsad.eskandary@meduniwien.ac.at (F.A.E.); 2Unidad de Investigación, Fundación Canaria Instituto de Investigación Sanitaria de Canarias (FIISC), Hospital Universitario de Canarias, 38320 Santa Cruz de Tenerife, Spain; rfreire@ull.edu.es; 3Instituto de Tecnologías Biomédicas, Centro de Investigaciones Biomédicas de Canarias, Facultad de Medicina, Universidad de La Laguna, Campus Ciencias de la Salud, 38200 Santa Cruz de Tenerife, Spain; 4Facultad de Ciencias de la Salud, Universidad Fernando Pessoa Canarias (UFP-C), 35450 Las Palmas de Gran Canaria, Spain; 5Center for Brain Research, Medical University of Vienna, 1090 Vienna, Austria; zsofia.hevesi@meduniwien.ac.at

**Keywords:** BK polyomavirus, urine sediment cells, qPCR, VP2, decoy cells

## Abstract

Polyomaviruses are widespread, with BK viruses being most common in humans who require immunosuppression due to allotransplantation. Infection with BK polyomavirus (BKV) may manifest as BK virus-associated nephropathy and hemorrhagic cystitis. Established diagnostic methods include the detection of polyomavirus in urine and blood by PCR and in tissue biopsies via immunohistochemistry. In this study, 79 patients with pathological renal retention parameters and acute kidney injury (AKI) were screened for BK polyomavirus replication by RNA extraction, reverse transcription, and virus-specific qPCR in urine sediment cells. A short fragment of the VP2 coding region was the target of qPCR amplification; patients with (n = 31) and without (n = 48) a history of renal transplantation were included. Urine sediment cell immunofluorescence staining for VP1 BK polyomavirus protein was performed using confocal microscopy. In 22 patients with acute renal injury, urinary sediment cells from 11 participants with kidney transplantation (KTX) and from 11 non-kidney transplanted patients (nonKTX) were positive for BK virus replication. BK virus copies were found more frequently in patients with AKI stage III (n = 14). Higher copy numbers were detected in KTX patients having experienced BK polyoma-nephropathy (BKPyVAN) in the past or diagnosed recently by histology (5.6 × 10^9^–3.1 × 10^10^). One patient developed BK viremia following delayed graft function (DGF) with BK virus-positive urine sediment. In nonKTX patients with BK copies, decoy cells were absent; however, positive staining of cells was found with epithelial morphology. Decoy cells were only found in KTX patients with BKPyVAN. In AKI, damage to the tubular epithelium itself may render the epithelial cells more permissive for polyoma replication. This non-invasive diagnostic approach to assess BK polyomavirus replication in urine sediment cells has the potential to identify KTX patients at risk for viremia and BKPyVAN during AKI. This method might serve as a valuable screening tool for close monitoring and tailored immunosuppression decisions.

## 1. Introduction

More than 90% of adults are infected with BK polyomavirus (BKV) and show humoral immunity to several genotypes [1]. Although it does not cause disease in healthy individuals, it can lead to graft failure and loss in kidney transplant (KTX) recipients. Data suggest that polyomavirus-associated nephropathy (PyVAN) in KTX patients arises from BKV infection of the donor organ. This leads to the question of which clinical conditions lead to active BKV replication in the non-transplanted kidney. Acute kidney injury (AKI) represents a condition of tubular cellular damage, which might make kidneys more susceptible to active infection and lead to BKV replication. 

A Swiss study of 400 healthy donors found a seroprevalence of 82% when virus-like particles were used as an antigen [2]. The proximal tubular cells of the kidney appear to be the reservoir for latent polyomavirus infection. For this reason, BK viruses can be detected in the fluid phase of urine of up to 15% of healthy individuals without evidence of disease-causing activity [3]. Infection latency can be reactivated in immunocompromised patients, such as those who are immunosuppressed after kidney [4] or bone marrow transplantation. In kidney transplant recipients, this virus can cause BK virus nephropathy (BKPyVAN) [5], whereas in bone marrow transplant recipients, it can lead to hemorrhagic cystitis [6]. In this gradual reactivation, viruria is followed by viremia [7], which is already considered by some authors as a sign of manifestation of polyomavirus-associated nephropathy (PyVAN). When such disease has manifested, graft function deteriorates in up to 22–67% of transplant patients with PyVAN, and the survival time of the transplant is reduced by half compared to the control group [8,9].

Until now, there is no pharmacotherapy available against BKPyVAN. The only attempt has been to modify immunosuppressive therapies [10,11], and with careful decision making, rejection episodes can still be kept low [12]. The clinical presentation is nonspecific, and urine evaluation for the presence of decoy cells and immunohistochemical examination of transplant biopsies are important diagnostic modalities in the differential diagnosis of elevated serum creatinine [4,13,14]. Recently, authors have tested other noninvasive diagnostic methods, such as the RTqPCR of urine sediment cells, to see if they are useful as biomarkers for BKPyVAN [15,16,17]. Immunostaining of decoy cells [18] or cytology using the Sternheim–Malbin method, which is quicker to perform, has also been shown to be effective [19]. It is assumed that the decoy cells are highly infected virus-replicating cells that originate from the tubular elements of the nephron and from further distally. Furthermore, it was assumed that they are generated by exfoliation as part of a lytic process in which non-enveloped viruses are passively released. However, there were recent reports that non-lytic egress was found for this non-enveloped virus [20]. It was already reported earlier that the non-enveloped Minute Virus in the mouse could be released from the host cell by lysosomal or endosomal vesicles [21]. Such a viral egress mode has also been reported for the release of hepatitis A viruses contained in vesicles covered by host membranes [22]. In this respect, it is important that a non-lytic release of BK polyomavirus from infected cells has been shown [23] and analysed in detail for BK polyomavirus recently [20]. A low-level subclinical polyomavirus replication rate and such release in patients with tubular damage but otherwise intact immune defenses could occur under conditions such as AKI.

In this study, we evaluated BK virus replication in urine sediment cells in the morning urine of patients with AKI and unexplained elevated urinary retention parameters. Urine cell pellets were analyzed for BK virus replication via RNA extraction and reverse transcription-PCR as well as via cell morphology and VP1 immunofluorescence staining. Both kidney allograft recipients (KTX) and non-transplant patients (nonKTX) were included in the analysis.

## 2. Results

A cohort of 79 patients was enrolled in the study between 2018 and 2021, including 31 KTX and 48 nonKTX patients (Figure 1). The average age of KTX patients and nonKTX patients was 55.9 and 58.4 years, respectively. AKI was diagnosed according to KDIGO criteria and classified into three stages. Patients with chronic kidney disease (CKD) were also classified according to KDIGO criteria. A total of 75 patients had elevated serum creatinine, and 4 had relevant proteinuria at enrollment. Comorbidities such as hypertension (n = 63), diabetes mellitus (n = 22), cardiovascular disease (n = 29), cerebrovascular disease (n = 10), and malignant disease (n = 15) were among the most important health problems of patients (Table 1).

The assay procedure used a commercial probe set from Applied Biosystems and, as shown in Figure 2, a short portion of the BK polyomavirus VP2 protein coding region, which is assumed to be conserved among all genotypes, was amplified in the quantitative analysis. For copy number evaluation, this amplicon was cloned into a plasmid that was included in a 10-fold dilution series as a standard curve.

In 22 patients with acute renal injury (AKI), urinary sediment cells from 11 participants with kidney transplantation and from 11 non-kidney transplanted patients were positive for BK virus replication. BK virus copies were found more frequently in patients with AKI stage III (n = 14) (Table 2). 

Among the 11 BKV-positive KTX patients, 3 of them had strikingly higher copy numbers than the other KTX patients (Figure 3). These three patients had biopsy evidence of BKPyVN either in the past or shortly before the urine sediment was collected. Two of them were found to be negative for decoy cell excretion, and the third, who had a strikingly higher copy number (by two magnitudes), had more than 40% decoy cells in the cytopreparation of the urine sediment, which stained positive for VP1 using mAb 4942. The patient’s VP1 cDNA was amplified by conventional PCR, and Sanger sequencing revealed genotype 1. All KTX and nonKTX patients who had BKV copies had AKI at the time of admission and measurement.

Out of the 79 included patients, 59 patients had a chronic kidney disease of stage 2 or higher. Among the 60 patients with AKI, including DGF, AKI stage III was the most common (n = 34). A total of 6 patients had AKI stage II, and 10 had AKI stage I. Ten patients post-KTX had a delayed graft function. One out of them was monitored over the first two weeks after transplantation when urine production had started (Figure 4). The BK virus copy number was increasing during that time in the urinary sediment cells. The patient later developed BK viremia but recovered after reducing immunosuppression.

### 2.1. BKV Spot Measurement in Urinary Sediment Cells of Patients at Time of Admission

BKV replication was detected more frequently in KTX patients (35.48%) compared to non-KTX patients (22.91%). To confirm the RTqPCR data, the amplicon was cloned into a plasmid and sequenced by Sanger sequencing.

### 2.2. Immunofluorescence Staining for VP1 in Urinary Cells

Confocal immunofluorescence was performed to investigate cell morphology among urinary excreted cells. As demonstrated in Figure 5, VP1 expression was found in cells of various morphology using the newly prepared rabbit anti-VP1 Ab (Figure 5A,B_1_,B_2_). In contrast, using the mAb 4942, no positive stains were found in nonKTX patients none of them showed decoy cell morphology in nonKTX patients by HE staining.

Positive staining cells (with rabbit anti-VP1 Ab) had typical urothelial or tubular epithelial morphology. In addition, some cells exhibited condensed chromatin and contained large degenerative vesicles (Figure 5B_1_,B_2_). In contrast, decoy cells positive for VP1 (with mAb 4942) were detected in 40% of total urinary cells in the patient with biopsy-verified BK polyomavirus nephropathy (Figure 6). These originate from proximal tubular cells, as indicated by AQP1 staining (Figure 6B). The copy number was, at this time point, 6.6 × 10^10^ (see Figure 3). Decoy cells exhibited strong cytosolic VP1 staining using mAb 4942, and the cytoplasmic membrane was ruptured in the decoy cells, as shown in the lower part of Figure 6A,C. VP1-positive small exosome-like particles spread near the lysed cells and adhered to the erythrocytes as a membrane rim. The erythrocytes are characterized by AQP1 staining (Figure 6B), and the VP1-positive granules adhering to the cytoplasmic membrane are shown in the VP1 image on the right (Figure 6C).

In the course of recovery from AKI, exfoliation of tubular epithelia and diapedesis of leukocytes is observed. In this line, we looked for BK copy numbers in the course of recovery. As shown in a 75-year-old female patient, the BK copy number increased as serum creatinine decreased and the patient recovered from delayed graft function (Figure 4). She developed BK viremia two months later, which reached a copy number of 22,000/mL in plasma at three months post-transplantation. This was considered indicative of the establishment of BKPyVNP, which led to a modification of the immunosuppressive regimen and subsequent recovery.

Finally, an estimate of graft survival was made for the KTX patients included in the study. As shown in Figure 7, BK-positive patients showed a lower graft survival rate than BK-negative patients.

## 3. Discussion

Screening for BKV in urine and blood is now a common practice in kidney transplant centers and is routinely performed by PCR testing for the detection of BKV DNA. In this study, the urine sediment of patients with and without kidney transplantation with AKI and pathological renal retention parameters was examined to detect actively transcribed BK virus using RNA. This work shows that even the silent presence and replication of polyomaviruses in the epithelia of the renal tract is promoted in AKI. However, the copy number never reaches such levels as in patients with BKPyVAN. Three KTX patients had very high copy numbers that differed from those of all other BKV-positive AKI patients. This was identified as bioptically verified BKPyVAN.

It seems important to point out that most polyoma replication was detected in patients with AKI stage III, suggesting that tubular epithelial injury itself might make epithelial cells more permissive for polyoma replication. As an additional factor, it could be considered that more tubular epithelial cells are excreted in AKI stage III than in stage I [24], increasing the chance of including BK virus-bearing cells in the qPCR measurement. This non-invasive method, such as the measurement of VP1 mRNA, was suggested by previous authors who showed that a strikingly high copy number of BKV VP1 in urine sediment indicates the presence of BKPyVAN and has a high diagnostic accuracy [17].

A high BKV concentration in the urine sediment due to AKI of the deceased donor organ could be a poor prognostic factor, making the kidney grafts susceptible to PVAN. In this context, donor viruria alone was shown to be a risk factor for BKV replication in the recipient after transplantation [25,26]. Therefore, it is of diagnostic importance to determine the BKV status of the donor, as the work of previous authors clearly demonstrates that BKV replication in the recipient originates from the donor [27]. This assumption is confirmed by our work with the 75-year-old study participant described in this paper. Furthermore, according to our data, AKI and DGF could by themselves be a risk factor favoring the spread of BKV to BKPyVAN in KTX patients.

In our study, decoy cells were not detected in the nonKTX group. This is confirmed by the morphology of VP1-positive cells in the immunostainings of these urine sediment cells. The VP1-positive cells in nonKTX patients did not have enlarged nuclei but showed large vesicles as a sign of degeneration (Figure 5B_1_,B_2_). Whether these structural changes were due to the AKI-associated cellular damage or due to the well-described virus-induced structural changes in the cytoskeleton [28,29] is undetermined. Rather, the percentages of dead cells and cell fragments were high in AKI patients, especially in stage III AKI. Conversely, decoy cells were only delineated in KTX patients.

In contrast to previous studies on the detection of actively translated BKV, which targeted the highly polymorphic VP1 protein [17], our work focused on the specified region of VP2. This part of the BKV genome does not appear to be as polymorphic, and PCR will not miss rare genotypes. However, it has recently been shown that this protein can also undergo mutations in rare cases [30]. Despite this potential ambiguity of their test, previous authors have already provided proof of concept by demonstrating high test specificity and sensitivity of over 90% for the diagnostics of BKPyVAN in their test setup [16].

It is likely that a renal allograft that has high viral transcripts in the cellular urine sediment at the stage of AKI or DGF after implantation represents an organ that could provide a breeding ground for the spread of polyomaviruses and the development of viremia or PVAN. This, together with other factors such as donor polyoma viruria, viremia, and high donor polyoma-specific antibodies [4], represents a new risk factor. Therefore, this method of polyomavirus detection in urinary sediment appears to be an appropriate tool for the stratification of transplant recipients for polyoma surveillance and immunosuppression right after transplantation. 

The rapid decline in viral replication after recovery from AKI in most of the patients despite immunosuppressive therapy is probably due to reduced shedding of injured tubule cells rather than the elimination of polyoma infection in proximal tubules. In this context, novel data were recently presented in which coordinated changes were observed in tubule cells experimentally infected with BKV [31]. These included genes for oxidative phosphorylation, the major signaling pathway assumed to be involved in the pathobiology of AKI. The damaged and weakened proximal tubule cells in AKI could open the door for polyomaviruses to replicate. This was illustrated by the experiment of previous authors that showed the downregulation of genes responsible for ribosomal proteins, glycolysis, and detoxification in proximal tubular cells with high viral expression [32].

The question of whether the amount of polyomavirus detected in urinary sediment cells contributes to the development of AKI is unanswered. However, the finding that the virus was absent in more than 50% of patients with AKI suggests that it was not causal in the development of AKI for our included patients but rather was related to cell damage and susceptibility to host cell permissiveness during AKI. 

Limitation of the study: the weakness of our study is the lack of prospective data with a representative patient sample. Conducting a prospective study could provide insights into the association between the measurement of viral replication in urinary sediment cells and the presence of polyomaviruria, viremia, and subsequent PVAN.

## 4. Materials and Methods

A total of 79 patients with pathological renal retention parameters, including kidney allograft recipients (KTX) and non-transplant (nonKTX) patients, were enrolled in this study. This study was approved by the Ethics Committee of the Medical University of Vienna under the number 1065/2021. Informed consent was obtained from each of the study participants.

Morning urine was collected into a sterile urinary container, and 8–10 mL urine was centrifuged at 2000× *g* for 10 min. From healthy donors who served as controls, 500 mL urine was centrifuged at 2000× *g* for 10 min. This high volume of urine was necessary to obtain a sufficient amount of cell sediment. The resultant pellet of urinary cells and debris was dissolved in 1000 μL TRIZOL (GIBCO BRL), which was frozen at −20 °C until further processing. For RNA isolation, the TRIZOL lysate was mixed with 200 μL chloroform and, after incubation for 15 min at room temperature, spun at 12,000× *g* for 10 min for phase separation. The aqueous phase was then mixed with 500 μL of isopropanol and spun at 12,000× *g* for 15 min to pellet the RNA. The RNA pellet was washed twice with 75% ethanol and briefly air-dried before being re-dissolved in nuclease-free water. 

### 4.1. Reverse Transcription

Reverse transcription was performed according to the protocol of superscript II (Invitrogen). Total RNA (800 ng) was mixed with hexamer random primers (Invitrogen, Waltham, MA, USA), dNTPs, first strand buffer, RNAse Out (Invitrogen, Waltham, MA, USA), and finally superscript II, which was incubated for 10 min at 25 °C for primer annealing and then for 50 min at 42° for first strand synthesis. The reaction was stopped by incubation at 75 °C for 10 min. The resultant cDNA was diluted 1:3 with TE buffer.

#### qPCR

For BK-polyomavirus replication among the sediment cells, 2 μL of the cDNA was combined with 8 μL of universal master mix (Applied Biosystems, Waltham, MA, USA) and BK-polyomavirus (Pa03453401_s1) probe set (Applied Biosystems, Waltham, MA, USA). The PCR reaction was run on a StepOne qPCR machine (Applied Biosystems, Waltham, MA, USA) recording 46 cycles for each run. For each urine sample, two technical replicates were included. A sample was considered positive when reaching the threshold level of amplification till cycle 39. A tenfold serial dilution of the plasmid that contained the sequence which was amplified by the BK-specific qPCR using the probe set Pa03453401_s1 was included as standard series. Copy numbers of all positive samples were related to this standard series. The generation of the plasmid used in the standard curve is described below. 

### 4.2. BK Polyoma Virus Copy Number Quantification

The amplicon of the qPCR obtained with the TaqMan probe set Pa03453401_s1 was used as template for cloning. This product was reacted with 1 μL of pENTR^TM^/D-TOPO vector (Invitrogen, Waltham, MA, USA) at room temperature for 7 min. The ligated plasmid was used for transforming One Shot^TM^ TOP10 *Escherichia coli* (*E. coli*) (Invitrogen, Waltham, MA, USA) using the heat shock method. Transformed *E. coli* was spread onto LB/kanamycin plates and incubated overnight at 37 °C. Outgrowing clones were tested by colony PCR, and two positive clones were selected for amplification in liquid culture. After 14 h bacterial culture, plasmids were isolated using PureLink^TM^ plasmid Miniprep kit (Invitrogen, Waltham, MA, USA). The plasmid insert had to be sequenced by Sanger sequencing. The sequence turned out to relate to VP2 (bp 1426–1487), a non-polymorphic region among BK genotypes. The plasmid DNA content was measured and entered into the copy number calculator (https://www.technologynetworks.com/tn/tools/copynumbercalculator, (accessed on 3 January 2023)). 

### 4.3. BK—VP1 PCR

As primer set for BK-VP1, the following forward primer ATGGCCCCAACCAAAAG and reverse primer TTAAAGCATTTTGGTTGCAATTG were used (Eurofins, Luxembourg). The cycling conditions were set to 94 °C for 4 min for initial denaturation; this was followed by 30 cycles of 94 °C for 30 s, 55 °C 30 s annealing, and 68 °C for 60 s synthesis. The PCR product was examined by Sanger sequencing.

### 4.4. Animal Care and Immunization

The animal study protocol (antibody production in rabbits) was approved by the Ethics Committee for Research and Welfare of Animals of the University of La Laguna (Comité de Ética de la Investigación y de Bienestar Animal, Universidad de La Laguna, CEIBA 2020-0391, 29 April 2020).

### 4.5. Rabbit Immunization

Antibody against VP1BK4 was raised by immunizing one female rabbit (*Oryctolagus cuniculus*) with the recombinant His-tagged proteins expressed in *E. coli* and purified with Ni-NTA agarose from Quiagen. Serum was collected after five injections of purified VP1 protein and adjuvant combined.

### 4.6. Immunofluorescence Staining of Urinary Sediment

An aliquot of cells was suspended in culture medium (RPMI supplemented with 10% calf serum), and 70 μL of the cell suspension was loaded into the funnel of a Shandon cytocentrifuge (Cytospin 3 Shandon, Runcorn, Cheshire, UK). The cyto-preparation was then air-dried and either stained with eosin/hematoxylin for immediate examination or frozen and wrapped in aluminum foil for further use, such as immunostaining.

For antibody incubation, cytopreparations were thawed while still stored in aluminum foil. After unwrapping, the slides were fixed in acetone for 5 min. A hydrophobic circle (with the Pap pen, Science Services) was drawn around the area where the cells were located. Twenty μL of PBS was applied to wet the cell preparation, and a BK polyomavirus-specific mAb (4942, Invitrogen, diluted 1:30) or the newly prepared rabbit anti-VP1 Ab (diluted 1:1000, described above) was applied to the slide. Antibody incubation was performed in a humidified chamber at room temperature (RT) for 1.5 h or overnight at 4 °C on a shaking platform. After a 10 min wash in PBS with constant agitation, the secondary antibody, either goat anti-mouse Alexaflour 594 or goat anti-rabbit Alexaflour 488 (each 1:700, in PBS), was applied and incubated again for 60 min in a humidified chamber on a shaking platform. For nuclear labeling, 40 μL of DAPI solution was applied and incubated for 3 min. After washing twice in PBS with constant stirring, the cytoslide was applied with Vectashield mounting medium for immunofluorescence (Vecotor Laboratories, Burlingham, CA, USA) and covered with a microscope glass coverslip (Chance Propper LTD, Warley, UK). Images were acquired using an Axiovert confocal microscope and processed using the Zen 3.4 (Zeiss, Jena, Germany) version of software and further illustrated using Photoshop CS6 (Adobe, San Jose, CA, USA).

Urine and blood analyses of BK- and JC-virus copy numbers were carried out at the Department of Laboratory Medicine of the Medical University of Vienna, which has an accredited and certified quality management system.

### 4.7. Statistical Analysis

We used the Mann–Whitney U-test to analyze differences between the study groups. Data were processed with GraphPad Prism 8, and *p*-values of less than 0.05 were considered significant. 

## 5. Conclusions

In AKI, damage to the tubular epithelium itself may render the epithelial cells susceptible to polyoma replication. The non-invasive diagnostic approach—a liquid biopsy—to measure BK polyomavirus replication in urinary sediment cells, targeting the VP2 transcript and cellular immunodetection of viruses in urinary sediment, has the potential to identify KTX patients at risk of developing BK viremia and BKPyVAN. Consistent with previous findings, this represents a useful screening tool for close monitoring and decision making for specific immunosuppression.

## Figures and Tables

**Figure 1 ijms-24-17511-f001:**
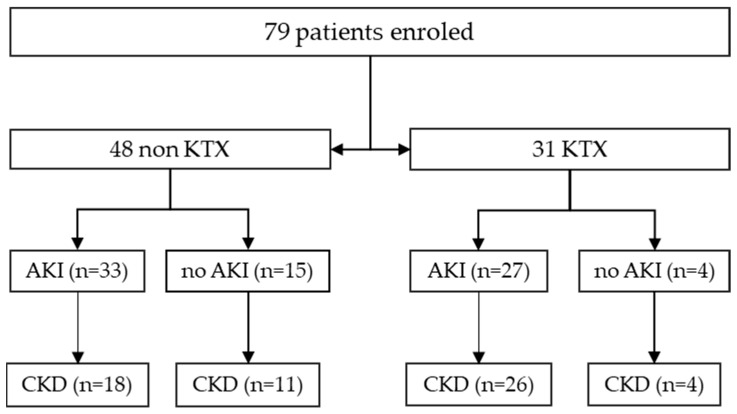
Enrolment algorithm for AKI and BK polyomavirus testing in urinary sediment cells. AKI: acute kidney injury, CKD: chronic kidney disease.

**Figure 2 ijms-24-17511-f002:**
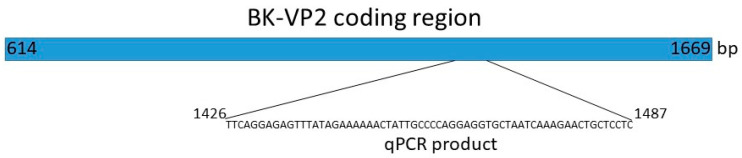
Schematic representation of the plasmid insert used in qPCR copy number measurement in urine sediment cells. This part of the VP2 gene represents the amplicon for the qPCR assay.

**Figure 3 ijms-24-17511-f003:**
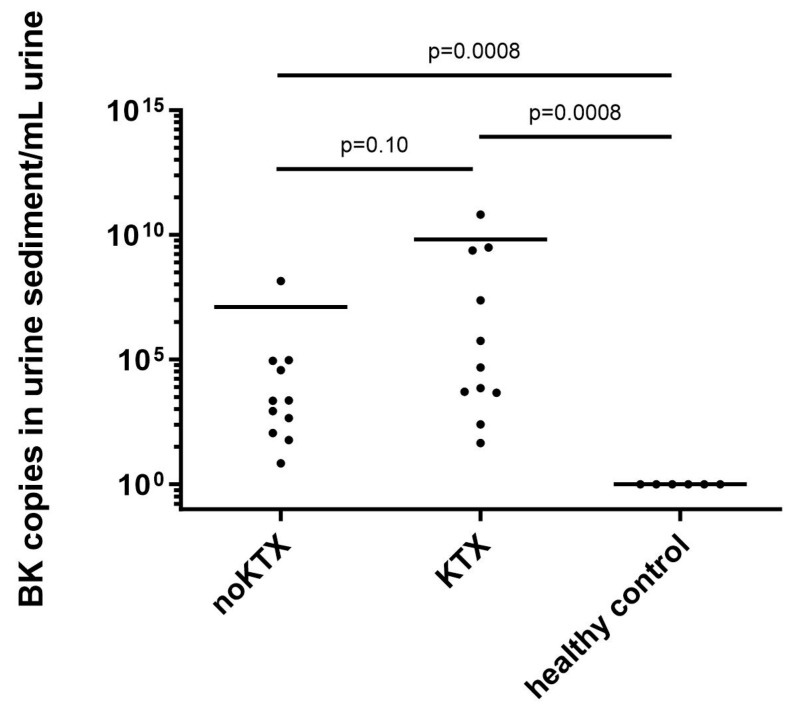
BK copy numbers in urine sediment cells. A total of 11 nonKTX patients and 11 KTX patients showed BK virus replication in urine sediment cells. In contrast, in 6 healthy control donors, no BK could be detected. Significant differences between the groups are indicated above the plots with *p*-values. KTX: kidney transplant, nonKTX: no kidney transplant.

**Figure 4 ijms-24-17511-f004:**
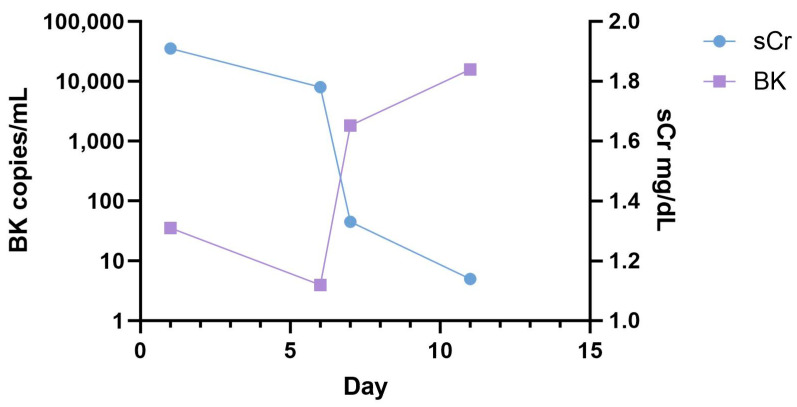
Course of the copy number of the BK virus in the urine cells. During DGF recovery, BK virus in urine sediment cells increased, peaking at 15,800 copies/mL urine. The patient developed BKPyVAN three months later. BK: BK polyomavirus copy number, sCr: serum creatinine.

**Figure 5 ijms-24-17511-f005:**
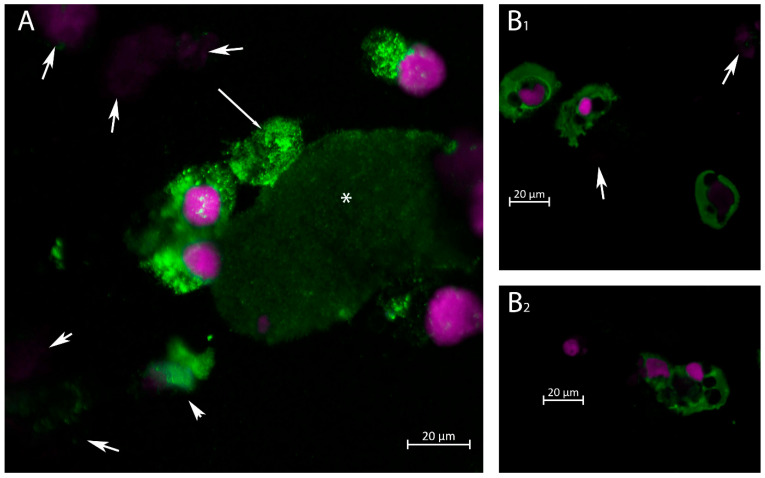
Immunofluorescence staining of urine sediment cells for VP1 (green) at the same time as positive qPCR examination in two patients (Patients (**A**,**B_1_**,**B_2_**)) with stage 3 AKI requiring renal replacement therapy. (**A**) Three VP1-positive cells (nucleus in purple) with tubular epithelial morphology, one of which with a long-tailed arrow represents a ghost cell with no nucleus. A patch is shown with weak positive staining (*), probably representing a cast with adherent BK virus. Cell debris is marked with arrows. The patient had 10^8^ copies of BK/mL urine in urine sediment determined by qPCR. (**B_1_**,**B_2_**) VP1-positive cells with urothelial morphology (green) and degenerative vesicles. Cell debris is indicated by short-tailed arrows.

**Figure 6 ijms-24-17511-f006:**
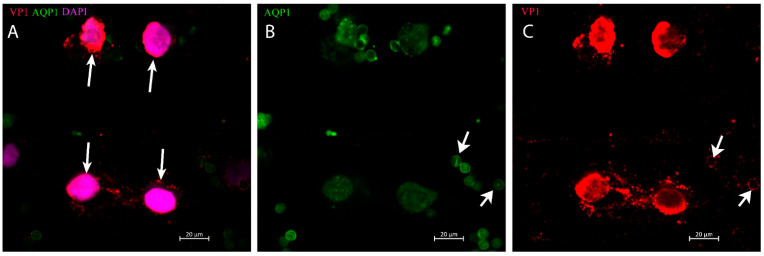
Four decoy cells stained with VP1, aquaporin 1 (AQP1), and DAPI in a KTX patient with BKPyVAN: (**A**) Decoy cells, indicated by a long-tailed arrow with enlarged nucleus (DAPI; pink) and VP1 positivity (red) showing cytoplasmic membrane damage. (**B**) Decoy cells stain positive for AQP1 (green) as indicator of proximal tubular origin together with erythrocytes also positive for AQP1. (**C**) Decoy cells with positive staining for VP1 (red) indicative of virus production and egress with two erythrocytes (indicated by a short-tailed arrow) with positive staining for VP1 at the membrane edge.

**Figure 7 ijms-24-17511-f007:**
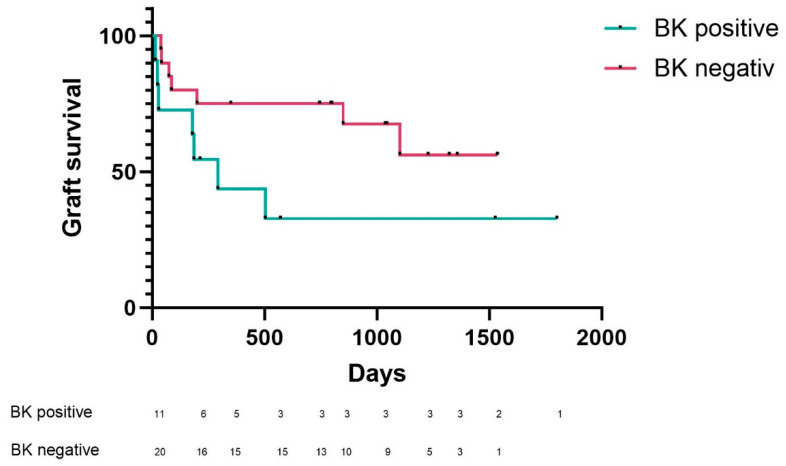
Analysis of graft survival in KTX patients. The graft survival in patients positive for BK in urinary sediment cells (n = 11) was remarkably reduced compared to BK-negative patients (n = 20).

**Table 1 ijms-24-17511-t001:** Demographics and characteristics of patients with (KTX) and without (nonKTX) kidney transplantation.

		KTX	nonKTX	Controls
**Characteristics**		31	48	6
**Gender**				
	Male	22	37	4
	Female	9	11	2
**Age ± SD**		55.97 ± 12.98	58.35 ± 19.24	46.17 ± 12.00
**Comorbidities**				
	Hypertension	28 (93.32%)	35 (72.91%)	
	Diabetes mellitus	11 (35.48%)	11 (22.91%)	
	Cardiovascular disease	15 (48.39%)	14 (29.16%)	
	Cerebrovascular disease	7 (22.58%)	3 (6.25%)	
	Malignancy	2 (66.45%)	13 (27.08%)	
	Immunosuppression	30 (96.77%)	8 (16.66%)	
**Chronic kidney disease**				
	stage 1	1 (3.23%)	19 (39.58%)	
	stage 2	3 (9.68%)	7 (14.58%)	
	stage 3	9 (29.03%)	11 (22.91%)	
	stage 4	4 (12.90%)	4 (8.33%)	
	stage 5	14 (45.16%)	7 (14.58%)	
**Acute Kidney injury**				
	No AKI	4 (12.90%)	15 (31.25%)	
	stage 1	5 (16.13%)	5 (10.41%)	
	stage 2	2 (6.45%)	4 (8.33%)	
	stage 3	10 (32.26%)	24 (50.00%)	
	post TX DGF	10 (32.26%)	n.a.	

Table 1 Demographics of study participants. DGF: delayed graft function, KTX: kidney transplant, nonKTX: no kidney transplant.

**Table 2 ijms-24-17511-t002:** Urinary sediment positive for BK Polyoma virus classified by stages of chronic kidney disease and acute kidney injury.

	KTX	nonKTX
**Number of participants with urinary**		
**sediment positive for polyoma**		
	n = 11	n = 11
**Chronic kidney disease**		
stage 1	1	5
stage 2	2	3
stage 3	4	3
stage 4	1	0
stage 5	3	0
**Acute kidney injury**		
stage 1	5	1
stage 2	1	1
stage 3	5	9
**Immunosuppression**		
No immunosuppression	n.a.	8
Triple IS	11	0
Dual IS	0	2
Mono IS	0	1
**KTX induction therapy (ATG)**	1	n.a.

Table 2 Stages of AKI and CKD and the immunosuppressive regimen in patients positive for BKV. ATG: antithymocyte globulin, KTX: kidney transplant, nonKTX: no kidney transplant.

## Data Availability

Data is contained within the article.

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
