# Peer review of "Acute Kidney Injury and BK Polyomavirus in Urine Sediment Cells"

_ijms, 2023, doi:10.3390/ijms242417511_

Round 1

Reviewer 1 Report

Comments and Suggestions for Authors

The manuscript investigates the replication of BK polyomavirus (BKV) in urine sediment cells of patients with acute kidney injury (AKI), particularly focusing on those with a history of kidney transplantation. The study involves 79 patients with pathological renal parameters and AKI, both with and without a history of renal transplantation. The researchers used RNA extraction, reverse transcription, and virus-specific qPCR to detect BKV replication in urine sediment cells. They targeted a specific region of the VP2 coding sequence for qPCR amplification. The results indicate that BKV replication is more frequent in patients with AKI stage III, and higher copy numbers are associated with kidney transplant recipients who have experienced BK polyoma-nephropathy. The study suggests that measuring BK polyomavirus replication in urine sediment cells could serve as a noninvasive diagnostic approach to identify kidney transplant patients at risk for complications such as viremia and BK polyoma-nephropathy when AKI is diagnosed. The manuscript also discusses the potential relationship between BK polyomavirus replication and acute kidney injury, highlighting the need for further research and prospective studies to validate the findings.

Although the manuscript is well-written, I recommend some minor revisions to enhance its clarity and overall quality.

1.      On page seven, Table 2 has an extra bracket after.

2.      On page 8, Figures 5 and 6 are in bold, whereas throughout the text figures and tables are not.

3.      On page 9, there is a spelling mistake in Figure 6 legend, and it should be corrected to "BK negative." Additionally, there is no indication of a sample size for the BK-negative group in the description box.

4.      On page 10, Figure 4B1 and B2 are recommended to be changed to Figure 4B1 and 4B2 for better clarity.

5.      Consider renaming the "Patient and Methods" section to a more general and encompassing "Materials and Methods," as it not only describes various methods used and patients enrolled but also includes information about animals used. It is advised to present the number of healthy donors. In the BK-VP1 PCR subsections, provide more information about the synthesis of primers: specify if they were synthesized elsewhere and, if so, name the commercial company. The title "Rabbit Immunization" contains an extra colon. Provide detailed information about the breed, age, sex, and the number of rabbits used in the experiment, as well as a more thorough description of the immunization process.

6.      Add a section on the methods used in the statistical analysis of the obtained data.

7.      Figure 3 doesn’t provide any information on the statistical significance of the data presented. Specify the method used to compare groups with different sample sizes.

Author Response

Please find our response in the word file attached.

Reviewer 2 Report

Comments and Suggestions for Authors

1. The control group is only mentioned in lines 94 and 203. What was the role of this group in your study? How many controls were included? ( I guess 6 cases, based on information in line 206) What were their characteristics? Were they comparable to the study group? Please, add some information on these issues.

2. Was informed consent obtained from the patients? If not, please, explain why.

3. Add clear Conclusions at the end of the article.

Author Response

(The authors gave the same response as above.)
